# OSMI-1 Enhances TRAIL-Induced Apoptosis through ER Stress and NF-κB Signaling in Colon Cancer Cells

**DOI:** 10.3390/ijms222011073

**Published:** 2021-10-14

**Authors:** Su-Jin Lee, Da-Eun Lee, Soo-Young Choi, Oh-Shin Kwon

**Affiliations:** 1BK21 FOUR KNU Creative BioResearch Group, School of Life Sciences, Kyungpook National University, Daegu 41566, Korea; neojove79@knu.ac.kr (S.-J.L.); starkr0@knu.ac.kr (D.-E.L.); 2Department of Biomedical Science and Research Institute of Bioscience and Biotechnology, Hallym University, Chuncheon 24252, Korea; sychoi@hallym.ac.kr

**Keywords:** O-GlcNAc transferase inhibitor, TRAIL, apoptosis, ER stress, NF-κB

## Abstract

Levels of O-GlcNAc transferase (OGT) and hyper-O-GlcNAcylation expression levels are associated with cancer pathogenesis. This study aimed to find conditions that maximize the therapeutic effect of cancer and minimize tissue damage by combining an OGT inhibitor (OSMI-1) and tumor necrosis factor-related apoptosis-inducing ligand (TRAIL). We found that OSMI-1 treatment in HCT116 human colon cancer cells has a potent synergistic effect on TRAIL-induced apoptosis signaling. Interestingly, OSMI-1 significantly increased TRAIL-mediated apoptosis by increasing the expression of the cell surface receptor DR5. ROS-induced endoplasmic reticulum (ER) stress by OSMI-1 not only upregulated CHOP-DR5 signaling but also activated Jun-N-terminal kinase (JNK), resulting in a decrease in Bcl2 and the release of cytochrome c from mitochondria. TRAIL induced the activation of NF-κB and played a role in resistance as an antiapoptotic factor. During this process, O-GlcNAcylation of IκB kinase (IKK) and IκBα degradation occurred, followed by translocation of p65 into the nucleus. However, combination treatment with OSMI-1 counteracted the effect of TRAIL-mediated NF-κB signaling, resulting in a more synergistic effect on apoptosis. Therefore, the combined treatment of OSMI-1 and TRAIL synergistically increased TRAIL-induced apoptosis through caspase-8 activation. Conclusively, OSMI-1 potentially sensitizes TRAIL-induced cell death in HCT116 cells through the blockade of NF-κB signaling and activation of apoptosis through ER stress response.

## 1. Introduction

Tumor necrosis factor (TNF)-related apoptosis-inducing ligand (TRAIL) is a type of cytokine known to induce apoptosis by binding to TRAIL-R1/DR4 and TRAIL-R2/DR5 [1,2]. TRAIL signaling is transmitted to the death-inducing signaling complex (DISC) present in the cell membrane and induces apoptosis by activating caspase-8 [3,4]. The extrinsic apoptosis pathway occurs through caspase-8 as it directly activates an effector, such as caspase-3 [5] and caspase-8 can cleave tBid, which interacts with Bax and Bak to induce mitochondrial damage [6,7]. Consequently, cytochrome c is released followed by the sequential activation of caspase-9 and caspase-3 [8,9]. In addition to activating apoptosis, TRAIL can induce nonapoptotic signaling, such as certain survival pathways that may promote resistance against TRAIL. Activation of proinflammatory pathways by TRAIL increases proliferation, migration, invasion and/or metastasis of cancer cells, primarily through nuclear factor-kappa B (NF-κB), PI3K, AKT and MAPK [10,11,12]. Recent studies have confirmed that NF-κB is often constitutively activated in human cancer cells [13].

NF-κB is a transcription factor that regulates the expression of genes involved in various functions including cell survival and immune response. Following a cytotoxic stimulus, such as LPS or TNF, IκB which forms a complex with NF-κB, is phosphorylated and degraded by IκB kinase (IKK) [14]. Consequently, free active NF-κB (p65) translocates to the nucleus to regulate the transcriptional activity of antiapoptotic proteins, such as Bcl2 and X-linked inhibitor of apoptosis (XIAP) [15]. In addition to nuclear translocation, the transcriptional activity of p65 is regulated by phosphorylation or acetylation modification [16]. Interestingly, it was demonstrated that the nuclear translocation of NF-κB (p65) was increased through O-GlcNAc modification [17]. O-GlcNAcylation modifies serine/threonine residues of intracellular proteins via O-GlcNAc transferase (OGT) or removes by O-GlcNAcase (OGA), resulting in diverse and dynamic modifications. As O-GlcNAcylation is increased in most malignancies, it is thought to be closely related to cancer progression, but the detailed molecular mechanisms have not yet been elucidated.

Extracellular environmental stress, such as reactive oxygen species (ROS), hypoxia, and nutrient deprivation, cause problems in the protein folding process and induce endoplasmic reticulum (ER) stress [18]. There are three major UPR signaling pathways that occur during ER stress, which include inositol-requiring enzyme 1α (IRE1α), protein kinase RNA (PKR)-like kinase (PERK) and Activating transcription factor 6 (ATF6) [19,20,21]. IRE1α produces X-box binding protein 1 (XBP1), which increases the folding ability of the ER and restores homeostasis to protect cells [22]. Moreover, it enhances apoptosis by activating Jun-N-terminal kinase (JNK) and p38 mitogen-activated protein kinase (p38 MAPK) [23]. Activation of PERK enhances the expression of the proapoptotic protein, CHOP by inducing the ATF4 transcription factor through phosphorylation of eukaryotic translation initiation factor 2α (eIF2α) [24]. CHOP regulates DR5 expression and several chemotherapeutic agents and proteasome inhibitors have recently been shown to enhance TRAIL-induced apoptosis by inducing ROS-dependent DR5 expression [25]. Unlike the other two sensors (PERK and IRE1α) ATF6 is both a sensor and a direct effector of UPR. During stress, ATF6 is translocated to the Golgi apparatus and is transformed into an active ATF6 p50 transcription factor, thereby enhancing the folding process in the ER.

Although TRAIL may be a relatively safe and promising death ligand in clinical applications, it has been reported that susceptibility to TRAIL-induced apoptosis depends on the type of cancer, and sensitive cancer cell lines may be resistant to TRAIL. Therefore, there is a need to develop TRAIL-based cancer therapeutics that are cancer-selective and exhibit novel mechanisms of action. In this study, we investigated the therapeutic efficacy of the combination of TRAIL and the OGT inhibitor, OSMI-1 for colorectal cancer. In the TRAIL-based cancer treatment strategy, OSMI-1’s role was to activate the ER stress response and blocks the NF-κB signal pathway. Moreover, the therapeutic agents were used at a low concentration to minimize side effects due to chemical toxicity. Consequently, this combination treatment was shown to synergistically increase anticancer activity by reducing cell proliferation and promoting apoptosis *in vitro* and *in vivo*.

## 2. Results

### 2.1. Combination Treatment with TRAIL and OSMI-1 in HCT116 and HepG2 Cells

We initially investigated the effect of combined OSMI-1 and TRAIL treatment on cell viability in HCT116 and HepG2 cells using the MTT assay. Cell viability was measured in response to a range of TRAIL concentrations alone or in combination with OSMI-1 (20 μM) for 24 h. As shown in Figure 1A, although the cell viability for TRAIL decreased in a concentration-dependent manner, the effect was insignificant at low concentrations in the range of 0.2–10 ng/mL. Treatment of HCT116 with increasing concentration of OSMI-1 for 24 h led to a dose dependent decrease in O-GlcNAc levels, as measured by western blot (Appendix A), without significant toxic effects in HCT116 treated with 5–40 μM (Appendix A). However, the combination with OSMI-1 significantly increased the concentration dependence of TRAIL and the proliferative activities were significantly inhibited in both cells. Changes in cell morphology were confirmed using an optical microscope (Figure 1B). No distinct morphological changes were observed in both cells when treated with a relatively low concentration (2 ng/mL) of TRAIL. However, it was confirmed that cells cotreated with OSMI-1 decreased the number of adherent cells and increased cell death. To determine whether the death of cells treated with TRAIL and OSMI-1 resulted from an activated apoptosis mechanism, the apoptosis rate was measured via flow cytometry after staining with Annexin V and PI. As shown in Figure 1C, OSMI-1 (20 μM) and TRAIL (2 ng/mL) treatment alone in both cells had a limited effect. In contrast, the apoptosis rate was significantly increased by the combination treatment. The apoptosis rate of HCT116 cells was 6.41 ± 1.25% in the control group, 22.36 ± 2.34% in the TRAIL group, 8.42 ± 1.37% in the OSMI-1 group, and 61.23 ± 5.32% in the combination group. In HepG2 cells, the degree of apoptosis was 2.43 ± 1.15% in the control group, 20.8 ± 3.27% in the TRAIL group, 3.26 ± 1.73% in the OSMI-1 group, and 36.8 ± 1.75% in the combination treatment. All these results suggested that cell death occurred through apoptosis and that OSMI-1 had a triggering effect in the apoptotic process. To confirm the process of apoptosis, caspase-3 levels were assessed in HCT116 and HepG2 cells cultured for 24 h (Figure 1D). TRAIL (2 ng/mL) treatment alone barely induced cleaved caspase-3 activation, whereas cotreatment with OSMI-1 dramatically enhanced the activation of capase-3. In addition, we determined the effect of OSMI-1 on TRAIL-induced apoptosis in SW620 cells and found that OSMI-1 enhanced TRAIL-induced activation of caspase-3 in these cells (Appendix A). To confirm that the combination treatment of TRAIL and OSMI-1 synergistically inhibited proliferation and cell growth, a cell colony formation assay was performed after cultivation for 14 days (Figure 1E). The rate of colony formation was significantly reduced after the combination treatment with TRAIL and OSMI-1 compared with TRAIL treatment alone. Altogether, these results indicate that the combination treatment for these cancer cells was much more effective and that OSMI-1 treatment sensitized both cells to TRAIL signaling, with higher sensitivity observed in HCT116 cells. Therefore, the remaining study focused on cell death resulting from combination treatment in HCT116 cancer cells.

### 2.2. TRAIL-Induced Apoptosis via Caspase-8 Is Enhanced by OSMI-1

To investigate the possible mechanism underlying the apoptosis effect of TRAIL and OSMI-1 in HCT116 cells, the levels of several apoptosis-related proteins were examined by western blot analysis. As shown in the panel of Figure 2A, total caspase-8 decreased in a dose-dependent manner during TRAIL treatment, Bid showed a decreasing trend, and no change was observed in cytochrome c. However, when cells were treated with OSMI-1, caspase-8 and Bid were not changed but cytochrome c increased in a dose-dependent manner. The results indicate that TRAIL-induced apoptosis occurs through the activation of caspase-8, whereas OSMI-1 treatment increases the release of the proapoptotic factor, cytochrome c, resulting from mitochondrial damage. However, the results presented in Appendix A indicate that the levels of cleaved caspase-3 were increased by OSMI-1 (40 µM). When 2 ng/mL TRAIL and 20 µM OSMI-1 were administered together, changes in the levels of total caspase-8 and Bid were further aggravated (Figure 2B). The level of the antiapoptotic protein, Bcl2, did not change during TRAIL treatment alone but it was significantly reduced after combination treatment. Apoptosis levels were also confirmed by measuring activated capase-3. Cleaved caspase-3 was hardly detectable following TRAIL treatment alone but was significantly increased when combined with OSMI-1 (Figure 2C). However, when Z-IETD-FMK, a caspase-8 inhibitor, was added to the combination, these effects disappeared. These results indicate that apoptotic cell death induced by the combination of TRAIL and OSMI-1 is caspase-8-dependent.

### 2.3. OSMI-1 Enhances TRAIL-Induced Apoptosis through ER Stress-Mediated CHOP/DR5

CHOP is a transcription factor belonging to the C/EBP family responsible for upregulation of DR5, and it is upregulated by ER stress and involved in ER-mediated apoptosis [25]. In this study, the association of OSMI-1 treatment with ER stress response in HCT116 cells was investigated. As shown in Figure 3A (left panel), the levels of several ER stress marker proteins including IRE1α, PERK and p-eIF2α increased proportionally with OSMI-1 treatment concentration. In the presence of OSMI-1, levels of CHOP and DR5 increased in a dose-dependent manner (right panel); however, CHOP and DR5 were not changed in the 2 ng/mL TRAIL-alone treatment. Further, we treated cells with CHOP siRNA to check the involvement of CHOP in the upregulation of DR5 (Figure 3B, in left panel). As expected, CHOP inhibition significantly inhibited upregulation of DR5, confirming that CHOP induction is required for upregulation of DR5 by OSMI-1. Moreover, siRNA-mediated silencing of CHOP completely blocked OSMI-1 induced cleavage caspase-3 (right panel). These data suggest that CHOP induction is one of the important mechanisms through which OSMI-1 enhances apoptosis in TRAIL-induced HCT116 cells. ER stress is induced by the accumulation of ROS which leads to mitochondrial dysfunction and apoptosis. In this study, we investigated intracellular ROS changes using DCFDA following treatment with OSMI-1 or TRAIL (Figure 3C). The results indicated that intracellular ROS was induced by 20 µM OSMI-1 but inhibited by pretreatment with N-acetylcysteine (NAC), a ROS scavenger. In contrast, it was difficult to detect ROS following treatment with 2 ng/mL TRAIL alone. Hence, we further evaluated the correlation of ROS generation with protein levels associated with ER stress response. ER receptors, including IRE1α and PERK, and their downstream effectors, p-elF2α, CHOP and DR5 were upregulated by OSMI-1 treatment. However, pretreatment with NAC significantly suppressed ER stress, reducing the levels of IRE1α, PERK, p-elF2α, CHOP and DR5 in OSMI-1 induced cells (Figure 3D). These results suggest that the ROS generated by OSMI-1 is directly related to ER stress response. Altogether, we demonstrated that OSMI-1 treatment induced ROS production and activated the ER stress, leading to upregulation of DR5 through CHOP.

### 2.4. ER Stress-Induced Apoptosis though JNK Activation by OSMI-1

We investigated the role of JNK in TRAIL-induced apoptosis following treatment with OSMI-1. To investigate OSMI-1 induced JNK activation, the expression levels of apoptosis-related proteins were determined via western blot analysis (Figure 4A). The activity of JNK along with cytochrome c increased in proportion to the concentration of OSMI-1, whereas the level of the antiapoptotic protein, Bcl2, gradually decreased. To determine whether the IRE1α receptor was responsible for the JNK effect, we performed a knockdown assay using siRNA against IRE1α. As expected, knockdown of IRE1α markedly attenuated the activation of JNK and returned the levels of cytochrome c and Bcl2 to that of the control group. We further evaluated whether OSMI-1 mediated JNK activation affected the apoptosis. p-JNK and subsequent p-c-Jun activation were increased upon OSMI-1 treatment and increased more significantly upon combination with TRAIL (Figure 4B). However, activation of apoptosis through the cleavage of caspase-3 and poly (ADP-ribose) polymerase (PARP) was not observed with OSMI-1 treatment alone, whereas it was evident following TRAIL treatment and more significant with the combination treatment. These results suggest that OSMI-1 induces JNK activation to enhance caspase-mediated apoptosis by TRAIL. For the combination treatment, we investigated the correlation between JNK activity and apoptotic factors relative to the concentration of TRAIL (Figure 4C). Relative to the concentration of TRAIL, p-JNK increased slightly, whereas cytochrome c and cleaved caspase-3 were significantly increased in OSMI-1 treated HCT116 cells. However, attenuation the activation of JNK using SP600125 returned the levels of cytochrome c and cleaved caspase-3 in HCT116 co treated with TRAIL and OSMI-1. These results suggest that the increase in apoptosis by the combination of TRAIL and OSMI-1 depends on JNK activation. Altogether, these data indicate that OSMI-1 played an important role in reducing Bcl2 and enhancing cytochrome c through IRE1α/JNK activation, resulting in increased TRAIL-mediated apoptotic cell death.

### 2.5. Modulation of TRAIL-Induced NF-κB Signaling by OSMI-1

Resistance to TRAIL-induced apoptosis in many cancer cells occurs through NF-κB activation. We investigated the effect of NF-κB signaling following treatment with TRAIL and OSMI-1 in HCT116 cells (Figure 5A). IKK was activated in a TRAIL concentration-dependent manner, although the level of IκB gradually decreased as soon as IκB were phosphorylated by IKK. Consequently, the level of phosphorylated p65 gradually increased. In contrast to TRAIL treatment alone, the activation of NF-κB upstream effectors, IKK and p65 was almost entirely blocked by cotreated OSMI-1. The level of Bcl2 also concomitantly increased in a TRAIL concentration-dependent manner but decreased significantly following the combination treatment. These results suggest that Bcl2, along with TRAIL-induced NF-κB activation, is associated with survival and is responsible for TRAIL sensitization. Next, we evaluated the involvement of O-GlcNAcylation in TRAIL-induced NF-κB signaling. O-GlcNAcylation levels increased gradually with TRAIL treatment and decreased significantly when TRAIL and OSMI-1 were administered together (right panel). Because IKKβ can be O-GlcNAcylated [26], we determined whether the activation of IKK by TRAIL or OSMI-1 treatment was related to O-GlcNAcylation. The levels of O-GlcNAcylated IKK were also analyzed by immunoprecipitation followed by Western blot analysis. As expected, the levels of O-GlcNAcylated IKK were increased by TRAIL treatment but disappeared by OSMI-1 combination treatment.

To analyze the effect of TRAIL and OSMI-1 on the nuclear translocation of p65, HCT116 cells were incubated with either TRAIL alone or in combination with OSMI-1 for 24 h and then confirmed by IF staining of p65 (Figure 5B). The results showed that there was a high nuclear translocation of p65 in TRAIL-treated cells, but OSMI-1 cotreated cells showed similar levels as the control. We further assessed the association of NF-κB with caspase cascade activity using Z-IETD-FMK treatment (Figure 5C). P-p65 levels were upregulated by TRAIL treatment alone but were clearly downregulated in following combination treatment with TRAIL and OSMI-1, whereas proapoptotic signals, such as cleaved caspase-3 and PARP, were detectable only in the combination treatment. However, Z-IETD-FMK treatment in TRAIL-induced HCT116 cells had no obvious effects on the phosphorylation of p65 compared with untreated cells. These results suggest that NF-κB inactivation induced by OSMI-1 significantly increased TRAIL-induced apoptosis, which occurs through activation of caspase-8. We confirmed the role of OSMI-1 in TRAIL-induced NF-κB activation and apoptosis pathways. The level of p65 activity increased with TRAIL treatment but was gradually downregulated in a dose-dependent manner when combined with OSMI-1 (Figure 5D). In contrast, the apoptosis signals, cytochrome c and cleaved caspase-3, were upregulated depending on the cotreatment concentration of OSMI-1. Moreover, we found that using p65 siRNA, p65 knockdown upregulated cleaved caspase-3 (Figure 5E). These results suggest that TRAIL activates NF-κB signaling for survival and growth, which causes resistance to apoptosis; however, when OSMI-1 is added, NF-κB signaling is inhibited, resulting in the promotion of apoptosis and suppression of tumor growth.

### 2.6. Combination Treatment Synergistically Enhances Anticancer Activity on HCT116 Xenograft in Nude Mice

Sensitization to TRAIL-induced apoptosis *in vitro* and *in vivo* (xenograft) in a large panel of cancer models has been well-established [27,28]. Based on the observed results *in vitro*, we next addressed the efficacy of TRAIL-based therapy *in vivo*. Using the subcutaneous HCT116 xenograft nude mouse model, we evaluated whether the combined treatment of TRAIL (500 μg/kg) and OSMI-1 (1 mg/kg) had a therapeutic effect. After administration of TRAIL or OSMI-1 alone or in combination for 28 days, the tumor volume over time was measured (Figure 6A in left panel) and the extracted tumors are shown in the middle panel. The tumor size in mice treated with TRAIL or OSMI-1 alone was slightly reduced compared with the control group but was significantly reduced (5-fold) in the TRAIL and OSMI-1 combination group (right panel). Western blot analysis revealed the levels of apoptosis proteins in the xenografts (Figure 6B). Compared with vehicle-treated xenograft mice, the levels of ER stress-related proteins, such as IRE1α, PERK, p-JNK, CHOP and DR5, were increased when either TRAIL or OSMI-1 was administered alone, whereas these effects were even greater for the combination treatment (in left panel). Note that the IRE1α /JNK/DR5 axis was specifically enhanced by OSMI-1 and further enhanced by the combination treatment. As expected, the levels of p-IκB, p-p65 and O-GlcNAcylation were significantly increased by TRAIL treatment, whereas only basal levels were found following OSMI-1 treatment or the combined treatment (in middle panel and Appendix A). Levels of Bid were decreased upon TRAIL treatment and decreased even more following the combined treatment with OSMI-1; however, the levels after OSMI-1 treatment alone were similar to that of the controls (right panel). The levels of cleaved caspase-3 and PARP were particularly upregulated in the combination treatment group, indicating that significant apoptotic death occurred in the combination treatment group. Altogether, these results are consistent with the results of previous cell experiments, wherein the compensatory action between TRAIL and OSMI-1 leads to decreased p65 signaling activity, and thus a synergistic increase in apoptosis via the ER stress pathway.

We performed immunohistochemistry (IHC) analysis to characterize the phenotype of the xenografts after each treatment (Figure 6C). Expression of the cell proliferation marker, Ki67, indicated that either TRAIL or OSMI-1 treatment alone had little or no effect on the reduction of tumor cell proliferation, whereas the combination treatment caused a dramatic decrease. Consistent with the results of western blot, the expression levels of cleaved caspase-3 showed the highest increase in the combination treatment, whereas p-IκB was increased in the TRAIL-treated group but absent in the combination treatment. In HCT116 cells, the combination treatment of TRAIL and OSMI-1 at relatively low concentrations synergistically induced apoptotic cell death. OSMI-1 upregulates CHOP and JNK through ER stress but negatively modulates NF-κB signaling through O-GlcNAcylation, resulting in apoptosis induction. Hence, we propose the underlying mechanism for this cell death wherein TRAIL activates the caspase-8 and NF-κB pathways, whereas OSMI-1 induces ER stress but blocks the NF-κB pathway. OSMI-1 significantly increased TRAIL-mediated apoptosis by increasing the expression of the cell-surface receptor DR5 by CHOP. OSMI-1 also activated JNK causing a decrease in Bcl2 and the release of cytochrome c from the mitochondria. When OSMI-1 was combined with TRAIL, there was a reduction in prosurvival NF-κB signaling (Figure 6D). The major downstream effectors of TRAIL included the activation of the caspase-signaling cascade and the NF-κB signaling pathway, whereas those for OSMI-1 were ER stress response elements and negatively modulated NF-κB signaling.

## 3. Discussion

Combination therapies consisting of anticancer agents with different molecular mechanisms of action generally induce synergistic effects compared with monotherapy. Recently, a number of combination therapies have been developed to enhance the efficacy of TRAIL and to overcome TRAIL resistance to improve the effect. In this study, we screened for potential therapeutic combinations involving OSMI-1 and TRAIL in colon cancer cell lines and showed that OSMI-1 enhances TRAIL-induced apoptosis. Furthermore, we demonstrated that the combination of TRAIL and OSMI-1 had a synergistic effect on growth inhibition in HCT116 xenograft tumors. The combination effect on cell death appeared to induce extrinsic pathways through TRAIL signaling and to sensitization through internal pathways by OSMI-1. Consequently, our mechanistic studies revealed that OSMI-1 enhances TRAIL-induced apoptosis by activation of the IRE1α /JNK/CHOP axis through ROS-ER stress as well as inhibition of the NF-κB signaling pathway.

When TRAIL binds to either DR4 or DR5, the DISC is formed in the cell, leading to caspase-8-dependent apoptosis. In addition to inducing apoptosis, TRAIL is known to promote cancer cell metastasis by activating NF-κB, a survival transcription factor [29]. Hence, we evaluated whether TRAIL treatment induces cell survival by activating the NF-κB pathway as well as caspase-8-dependent apoptosis. In HCT116 cells treated with a relatively low concentration (2 ng/mL) of TRAIL, caspase-8 was activated; however, cytochrome c release or caspase-3 activation was hardly confirmed. On the other hand, TRAIL treatment upregulated IKK and induced IκB degradation and p65 nuclear localization [30,31] Consequently, TRAIL treatment increased the activity of NF-κB, whereas it induced caspase-8 activation. These results suggest that activation of NF-κB acts as an apoptosis resistance mechanism [32]. Moreover, TRAIL treatment remarkably increased protein O-GlcNAcylation. Thus, we speculated that apoptotic signaling was probably attenuated by upregulating O-GlcNAcylation. Overall, apoptotic cell death hardly occurred in TRAIL treatment alone at a low concentration. These results suggest that activation of NF-κB acts as an effect that limits the ability of TRAIL to induced apoptosis.

Interestingly, treatment with the combination of TRAIL and OSMI-1 synergistically promoted ER stress, resulting in a dramatic increase in apoptosis in HCT116 cells. Evidence suggests an interrelation between ER stress and NAC is generation. Recently, ROS has been reported to play a critical role in both DR5 upregulation and TRAIL-induced apoptosis [33,34]. Therefore, we investigated whether OSMI-1 was involved in ROS production and ER stress response, resulting in sensitized TRAIL-induced apoptosis. Treatment with OSMI-1 induced ROS production, whereas pretreatment with NAC, significantly reduced OSMI-1-induced ER stress and CHOP-dependent DR5 expression. Moreover, apoptosis induced by the combination treatment of TRAIL and OSMI-1 was significantly attenuated by NAC. These results indicate that ROS acts as an upstream signaling molecule initiating OSMI-1 induced ER stress. Nevertheless, further studies are warranted to elucidate the function of OSMI-1 as a link to the ROS/ER-stress axis.

It is well known that alterations in redox state or ROS generation directly or indirectly affect ER homeostasis and protein folding processes. A recent report suggested that ROS may stimulate IRE1α signaling to induce apoptosis [35]. Therefore, in this study, we focused on the signaling pathway of ER stress induced by OSMI-1. We confirmed that OSMI-1 induces the activation of two important signaling pathways, PERK-elF2α and IRE1α-JNK, through ER stress (Figure 3 and Figure 4). The PERK/eIF2α signaling pathway primarily drives CHOP and DR5 induction but also occurs through the IRE1α pathway. It has been reported that CHOP is a typical ER stress regulatory protein involved in apoptosis, and ROS generation is a major target for triggering and amplifying TRAIL-dependent apoptosis through DR5 expression. Thus, it appears that ROS induced by OSMI-1 upregulates IRE1α and CHOP/DR5 expression. Indeed, OSMI-1 increased the expression of CHOP and the expression of DR5 was upregulated, but Bcl2 was downregulated. Therefore, OSMI-1 induced ER stress response leads to caspase-8 and Bid activation, resulting in TRAIL induced apoptotic cell death. We showed here for the first time that OSMI-1 effectively sensitizes HCT116 cells to TRAIL-induced apoptosis through DR5 upregulation. JNK is a classical apoptotic protein that is activated by IRE1α during ER stress [36,37,38]. Singh and colleagues reported that OSMI-1 induces apoptosis by ROS-dependent activation of JNK. In our study, OSMI-1 treatment increased protein levels for IRE1α and enhanced JNK phosphorylation. We confirmed that JNK activation is dependent on IRE1α by treatment with IRE1α siRNA. Combination treatment of OSMI-1 with TRAIL was shown to significantly enhance caspase-dependent apoptosis in HCT116 cells through JNK activation along with DR5 upregulation. Furthermore, our results showed that mitochondrial damage may be regulated by JNK activation because pretreatment of cells with the JNK inhibitor, SP600125, significantly reduced OSMI-1 induced cytochrome c release. In conclusion, OSMI-1 upregulated the TRAIL death receptor by activating IRE1α/JNK/CHOP signaling.

Inhibition of the NF-κB pathway can enhance the efficacy of cancer treatment and it has recently been reported that OGT is a key regulator of NF-κB activation [39,40]. OGT inhibitors inhibit the phosphorylation of IKKα/β and p65, inactivating NF-κB, a known survival-promoting factor. In this study, it was confirmed that TRAIL treatment in HCT116 cells resulted in upregulation of NF-κB activity, whereas OSMI-1 treatment resulted in a decrease of Bcl2 and inactivation of the NF-κB signaling pathway. TRAIL-induced p65 nuclear translocation is a well-known cause of increased NF-κB activity; however, treatment with OSMI-1 resulted inactivation of the NF-κB mechanism. These results suggest that TRAIL increases survival signaling through NF-κB but is counteracted by OSMI-1, ultimately resulting in a significant induction of apoptosis. Consequently, the combination treatment resulted in a more dramatic increase in apoptosis and a further decrease in tumor growth. 

Studies indicate that excessive O-GlcNAc modification occurs in many cancer cells, which seems to be directly related to cell survival in various cellular stress environments, including ER stress. Previous studies, including ours, revealed that O-GlcNAcylation enhances the activity of NF-κB in various cancer cells. Lowering hyper-O-GlcNAcylation in pancreatic cancer cells reduces IKKβ expression and attenuates p65-activated phosphorylation, nuclear translocation, and NF-κB transcriptional activity. Furthermore, a recent study revealed that O-GlcNAc modification of p65 promotes acetylation, which in turn regulates the activity of NF-κB transcription. However, the mechanism by which O-GlcNAcylation activates NF-κB signaling via TRAIL-induced phosphorylation has not been fully elucidated. In this study, we confirmed that NF-κB signaling was directly regulated through O-GlcNAc modification. We showed through IP analysis that O-GlcNAc modification occurs in IKKβ during TRAIL-induced HCT116 apoptosis (Figure 5). The induction of TRAIL resulted in increased O-GlcNAcylation and activation of the NF-κB signal pathway. However, we confirmed that the elimination of O-GlcNAc from IKK by OGT inhibitor treatment blocked the activity of IKK and the downstream signaling pathway, resulting in a decrease in p65 nuclear translocation. Therefore, although more studies are needed, it is evident that increased O-GlcNAcylation in cancer cells is directly related to the activation of the NF-κB signaling pathway through the activation of IKK. The results of this study suggest that blockade of O-GlcNAcylation signaling by OSMI-1 directly effects cell survival and sensitivity to trace signals, thereby promoting apoptosis.

We showed here that OSMI-1 sensitizes TRAIL induced cell death pathway via a parallel mechanism of ER stress response and NF-κB signaling. TRAIL induces apoptosis through caspase-8 activation, whereas OSMI-1 activates CHOP and JNK through ER stress response. Thus, DR5 was upregulated and cytochrome c was released, which in turn increased apoptosis. Conversely, the NF-κB signaling pathway involved in cell survival was positively upregulated by TRAIL leading to TRAIL resistance, whereas it was negatively regulated through lack of O-GlcNAcylation by OSMI-1. These results suggest that NF-κB crosstalk significantly contributes to the decision between cell survival and death. Consequently, our findings provide evidence that OSMI-1 potentially sensitizes TRAIL-induced cell death in HCT116 cancer cells by blocking the NF-κB signaling pathway and inducing apoptosis through DR5 and JNK activation. Therefore, the combination of OSMI-1 and TRAIL may provide a promising therapeutic strategy for the treatment of various cancer cells.

## 4. Materials and Methods

### 4.1. Cell Culture and Transfection

HCT116 and HepG2 cells were obtained from the Korean Cell Line Bank (Seoul, Korea) and then maintained in DMEM media containing 10% fetal bovine serum and 1% penicillin. Cells were cultured at 37 °C in a humidified 5% CO_2_ incubator. The recombinant human TRAIL and Z-IETD-FMK were obtained from R&D systems (Minneapolis, MN, USA). The cancer cells were treated with 0.2–10 ng/mL final concentration of TRAIL. CHOP siRNA construct was as follows: 5′-GCCUGGUAUGAGGACCUGC-3′. Control, IRE1α, and p65 siRNA (Santa Cruz, Dallas, TX, USA) were transfected into HCT116 cells using Lipofectamine RNAi MAX (Invitrogen, Waltham, MA, USA). Cells were lysed in NP40 buffer and western blot procedure was performed. Then, the cells were incubated overnight with primary antibodies in TBST. Anti-IRE1α and p65 were purchased from Cell Signaling Technology (Danvers, MA, USA).

### 4.2. Preparation of Cell Lysates and Western Blot Analysis

HCT116 and HepG2 cell samples were lysed with NP40 assay buffer containing 1× PBS, 1% NP40, 1 mM EDTA, and protease inhibitor cocktail tablets (Roche, Mannheim, Germany). Proteins were separated on 8–15% SDS-PAGE gels and transferred to nitrocellulose membranes (GE Healthcare Life science, Boston, MA, USA). The membranes were blocked with 5% skim milk at room temperature for 1 h and then incubated overnight with primary antibodies at 4 °C. Anti-O-GlcNAc was purchased from Sigma-Aldrich (St Louis, MO, USA). Anti-PARP was purchased from Invitrogen (Waltham, MA, USA). Anti-total caspase-3, cleaved caspase-3, IRE1α, p-p65, p65, p-IKKα/β, IKKα/β, p-IκB and p-eIF2α were purchased from Cell Signaling Technology. Anti-OGT, PERK, Bcl2, Bax, cytochrom c and β-actin were purchased from Santa Cruz. Membranes were washed with TBST and incubated with antimouse or antirabbit horseradish peroxidase (HRP)-conjugated IgG secondary antibody for 30 min. The signal was visualized using an ECL Plus Detection System (GE Healthcare Life Sciences) and the bands were quantified using ImageJ software (NIH, United States).

### 4.3. Cell Viability Assay

The cells were seeded at a density of 1 × 10^4^ cells/well for 1 day in 96-well cell culture plates. Cell viability was measured using a 3-(4,5-dimethylthiazol-2-yl)-2,5-diphenyl-tetrazolium bromide (MTT) assay. After 24 h of TRAIL exposure, 2 mg/mL MTT (Sigma-Aldrich) solution was added to each well and incubated at 37 °C with 5% CO_2_ for 4 h. The MTT solution was discarded and 100 μL of dimethyl sulfoxide (DMSO) was added to each well. The OD of the wells was measured at 580 nm using a spectrophotometric microplate reader (Molecular Devices, San Jose, CA, USA). The experiments were performed in triplicates and expressed as the percentage of cell viability.

### 4.4. Annexin V/Fluorescein Isothiocyanate (FITC) Flow Cytometric Assay

The percentage of apoptosis by TRAIL and OSMI-1 was measured using an annexin V fluorescein isothiocyanate (FITC) and propidium iodide (PI) apoptosis detection kit (Thermo Fisher Scientific, MA, USA). Briefly, each cell line was seeded at 2 × 10^5^ cell/mL in cell culture dishes. Cells were treated with 2 ng/mL TRAIL with or without 20 μM OSMI-1 and incubated for 24 h. Then, the cells were harvested with trypsin, washed once with PBS and immediately suspended in 100 μL of 1× Annexin V binding buffer. FITC Annexin V (5 μL) was added and allowed to react at room temperature for 15 min and 0.1 μL of PI were added to each tube. The mixture was placed on ice (4 °C) in the dark. Finally, 400 μL of 1× Annexin V binding buffer were added to each tube and the samples were analyzed via fluorescence-activated cell sorting (FACS) (BD Biosciences, CA, USA).

### 4.5. Measurement of the Accumulation of ROS

Cells were loaded with 10 μM dichlorofluorescein diacetate (DCFDA) (Invitrogen) and then incubated for 30 min at 37 °C after washing with PBS buffer. Intracellular accumulation was examined under a fluorescence microscope and measured via flow cytometry.

### 4.6. Immunoprecipitation (IP)

Reciprocal IP was done using HCT116 lysates and antibody against IKKβ using the Dynabeads^TM^ Protein G IP kit (Thermo Fisher Scientific). The immunoprecipitated protein was resolved from immune complexes precipitated with protein G by boiling in 2× sample buffer and then subjected to SDS-PAGE and western blot analysis.

### 4.7. Colony Formation Assay

A total of 1000 HCT116 and HepG2 cells were seeded into six well plates and the medium were changed every 4 days. The cells were grown for 2 weeks to form colonies at 37 °C and then washed twice with PBS. The colonies were fixed with 100% methanol and stained with 0.5% crystal violet. The colonies that had formed were counted and expressed as percent control under an optical microscope. All assays were performed thrice.

### 4.8. Tumor Implantation and Growth

This study was approved by the Animal Care and Use Committee of the Kyungpook National University, Korea. Five-week-old female BALB/c-Foxn1nu/ArcGem nude mice were purchased from GEM Biosciences Inc. (Cheongju, Republic of Korea). HCT116 xenograft models were injected subcutaneously with 5 × 10^6^ cells in the flanks of each mouse. After 7 days, tumor-bearing mice (tumor volume approximately 90–110 mm^3^) were randomized into various treatment groups (*n*  =  4/group) as follows: Group 1—mice were administered DMSO as a vehicle control; Group 2—mice were administered TRAIL (500 μg/kg/daily, Intraperitoneal); Group 3—mice were administered OSMI-1(1 mg/kg/daily, Intravenous); Group 4—mice were coadministered TRAIL and OSMI-1 for 21 consecutive days. Tumors were measured with calipers according to a score sheet that recorded tumor size calculated by the following formula: length × width^2^ × 0.5 (mm^3^). Animals were also weighed weekly and monitored for general health status and signs of possible toxicity from the treatment. The tumors were fixed in 4% paraformaldehyde for immunohistochemistry (IHC) and western blot analysis.

### 4.9. Immunohistochemistry

Xenograft tumor tissues were extracted from xenograft mice. The tumors were fixed in 4% paraformaldehyde and embedded in paraffin. Sections were then deparaffinized and boiled in citrate buffer (pH 6.0) for blocking endogenous peroxidase. After cooling at room temperature, samples were incubated in 5% BSA containing 10% normal goat serum for 1 h. The samples were incubated with anti-cleaved capase-3, Ki67, p-IκB antibody (1:200 dilution) at 4 °C for histological analysis. Samples were incubated with the appropriate secondary antibodies. The reaction products were visualized with 3, 3-diaminobenzidine (DAB) reagent.

### 4.10. Statistical Analysis

All experiments were expressed as the mean ± SEM. Two sample t-tests were performed. All data were repeated at least 3–5 times. Statistical analysis was done using Student’s t-test for comparison between the groups and *p*-values <0.05, <0.01 or <0.005 were considered statistically significant. All data analyses were performed using SPSS software.

## Figures and Tables

**Figure 1 ijms-22-11073-f001:**
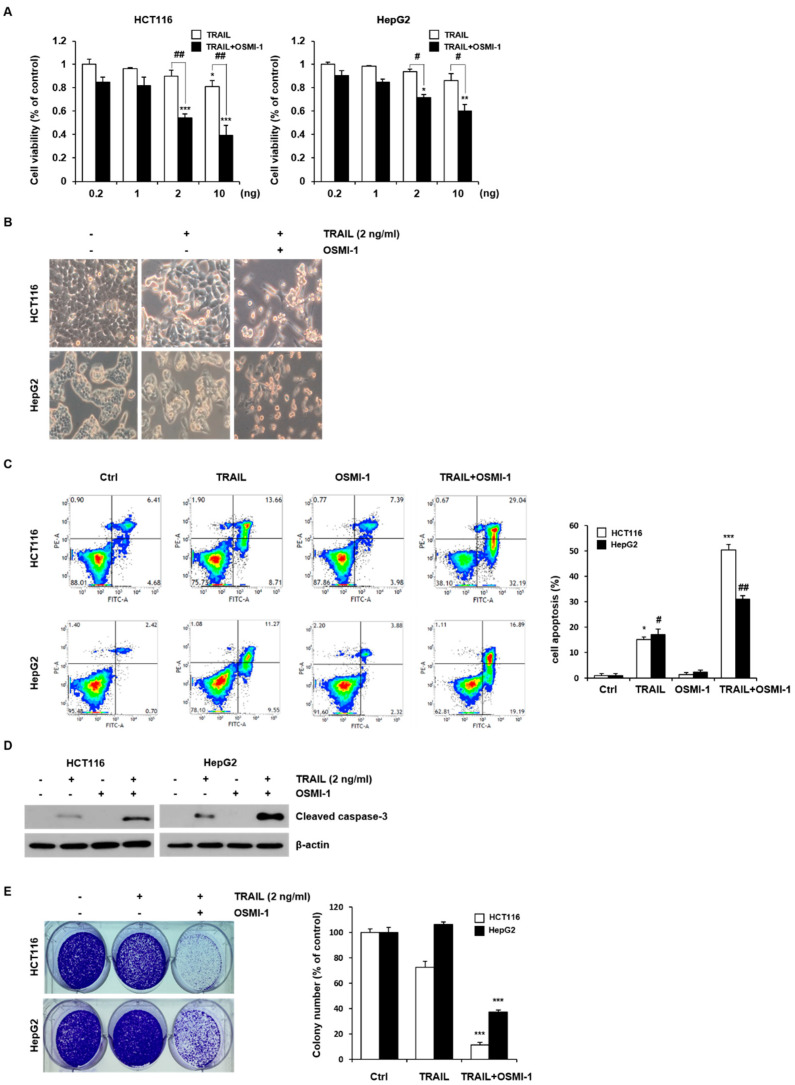
Combination treatment of TRAIL and OSMI-1 in HCT116 and HepG2 cells. (**A**) Two cancer cell lines (HCT116 and HepG2) were treated with various concentrations of TRAIL (0.2–10 ng/mL) in the absence or presence of OSMI-1 (20 µM) for 24 h, and cell viability was determined via MTT assay. Data are expressed as the mean ± SEM for 3–4 independent experiments; * *p* < 0.05, ** *p* < 0.01, *** *p* < 0.005 vs. control. ^#^
*p* < 0.05, ^##^
*p* < 0.01 vs. trail alone. (**B**) Cells were treated with 2 ng/mL TRAIL alone or in combination with 20 µM OSMI-1. The morphology of the cells was evaluated by interference light microscopy (magnification, ×200). (**C**) Apoptosis assay using flow cytometry after a 24 h treatment with TRAIL or OSMI-1 alone or the combination of both. Quantitative data for three independent flow cytometry experiments. * *p* < 0.05, *** *p* < 0.005 vs. control HCT116 cells. ^#^ *p* < 0.05, ^##^
*p* < 0.01 vs. control HepG2 cells. (**D**) Total cell lysates were collected from HCT116 and HepG2 cells treated with 2 ng/mL TRAIL or 20 µM OSMI-1 alone or in combination for 24 h. Caspase-3 activity was measured via western blot analysis. Actin was used as a loading control for each lane. (**E**) A colony formation assay was used to determine the clonogenic capacity of HCT116 or HepG2 cells in the absence or presence of TRAIL (2 ng/mL) alone or in combination with OSMI-1 (20 µM). After cultivation for 14 days, plates were stained with crystal violet. Colonies for three independent experiments were counted. The graphs on the right show the mean number of colonies as a relative value (%). *** *p* < 0.005 vs. control cells.

**Figure 2 ijms-22-11073-f002:**
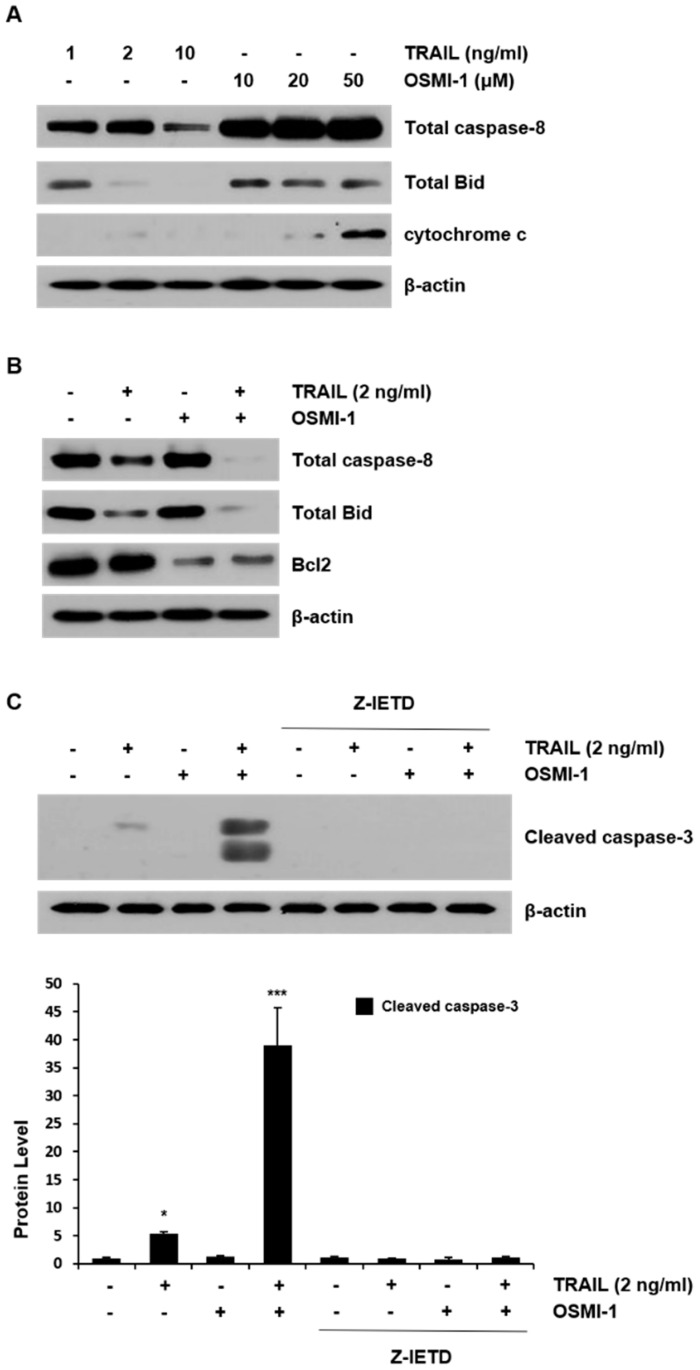
OSMI-1 enhances TRAIL-induced apoptosis in HCT116 cells. (**A**) HCT116 cells were treated with TRAIL (1, 2 and 10 ng/mL) or OSMI-1 (10, 20 and 50 µM) for 24 h. Whole-cell extracts were then prepared and the levels of caspase-8, Bid and cytochrome c were determined by western blot analysis. (**B**) Total cell lysates were collected from HCT116 cells treated with 2 ng/mL TRAIL or 20 µM OSMI-1 alone or in combination for 24 h. Caspase-8, Bid and Bcl2 levels were measured by western blot analysis. Actin was used as a loading control for each lane. (**C**) Cells treated with TRAIL and/or OSMI-1 or in combination with or without 10 µM Z-IETD-FMK were analyzed using western blot analysis. Data are expressed as the mean ± SEM for 3–4 independent experiments; * *p* < 0.05, *** *p* < 0.005 vs. control.

**Figure 3 ijms-22-11073-f003:**
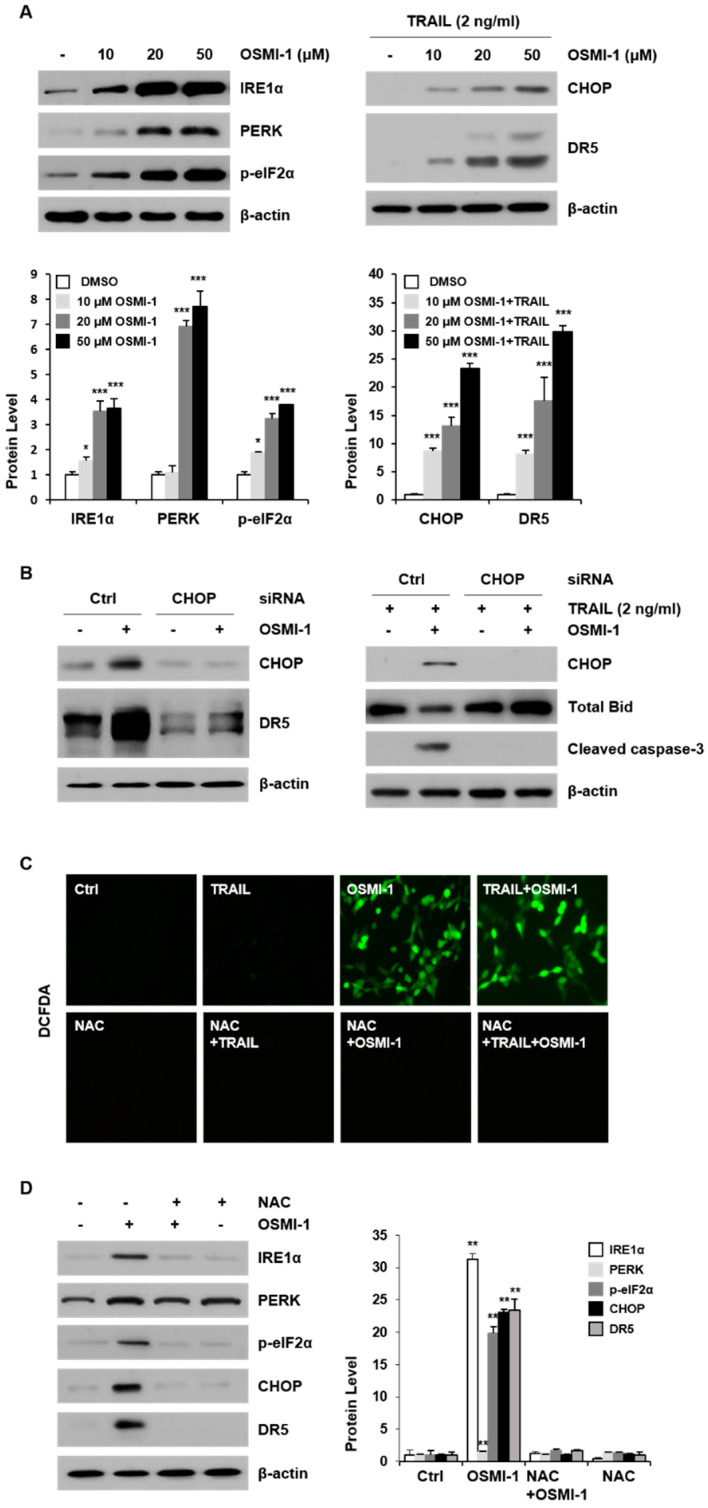
OSMI-1 induces ER stress signaling in HCT116 cells. (**A**) Cells were treated with the indicated concentrations of OSMI-1 for 24 h. After treatment, the expression levels of ER stress markers, IRE1α, PERK and p-eIF2α were determined via western blot analysis (left panel). Cells were treated with 2 ng/mL TRAIL for 24 h in the absence or presence of various concentration of OSMI-1. Total cell lysates were subjected to western blot analysis with the indicated antibodies against CHOP and DR5 (right panel). Data are expressed as the mean ± SEM for 3–4 independent experiments; * *p* < 0.05, *** *p* < 0.005 vs. control. (**B**) Cells were transfected with CHOP siRNA or control siRNA for 24 h and then treated with OSMI-1 for an additional 24 h. The protein levels of CHOP and DR5 were determined via western blot analysis (left panel). Likewise, cells were transfected with CHOP siRNA for 24 h and then treated with TRAIL in the absence or presence OSMI-1. Cell lysates were analyzed via western blot analysis using antibodies against CHOP, Bid and cleaved caspase-3 (right panel). (**C**) Cells were either untreated or pretreated with NAC (5 mM) for 1 h and then treated with OSMI-1 (20 µM) for 24 h. To measure cellular ROS generation, immunofluorescence of DCFDA was determined via confocal laser scanning microscopy. (**D**) Cells were pretreated with NAC (5 mM) for 1 h and then treated with OSMI-1 (20 µM) for 24 h. The expression levels of ER stress markers (IRE1α and PERK) and downstream effectors (p-eIF2α, CHOP and DR5) were determined via western blot analysis. Data are expressed as the mean ± SEM of 3–4 independent experiments; ** *p* < 0.01 vs. control.

**Figure 4 ijms-22-11073-f004:**
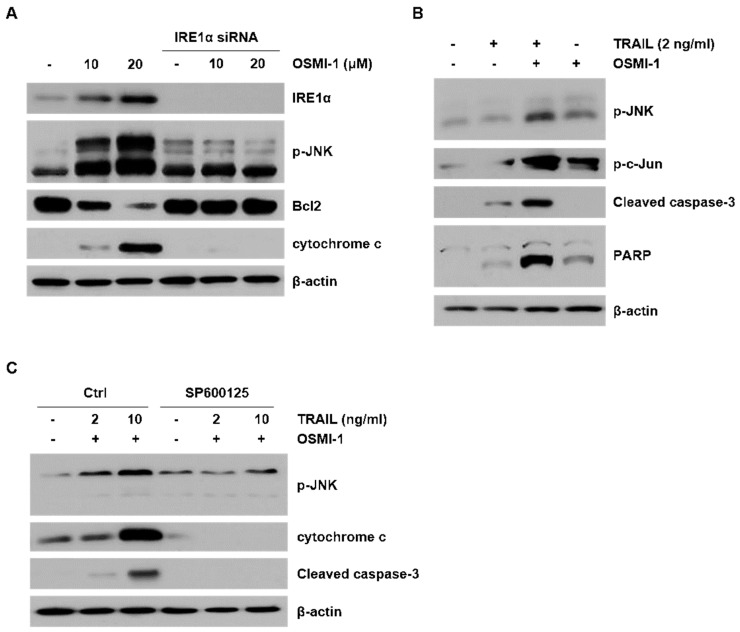
OSMI-1 enhances TRAIL-induced apoptosis through JNK. (**A**) Cells were transfected with control or IRE1α siRNA and then treated with various concentrations of OSMI-1 for 24 h. Cell extracts were analyzed via western blot for IRE1α, p-JNK, Bcl2 and cytochrome c. (**B**) HCT116 cells were treated with TRAIL (2 ng/mL) or OSMI-1 (20 µM) alone or in combination for 24 h. Western blot analysis was performed to determine the levels of p-JNK, p-c-Jun, cleaved caspase-3 and PARP. (**C**) Cells were pretreated with or without SP600125 (JNK inhibitor) for 1 h and then cotreated with TRAIL (2 or 10 ng/mL) and OSMI-1 (20 µM) for 24 h. Expression levels of p-JNK, cytochrome c and cleaved caspase-3 were determined via western blot analysis.

**Figure 5 ijms-22-11073-f005:**
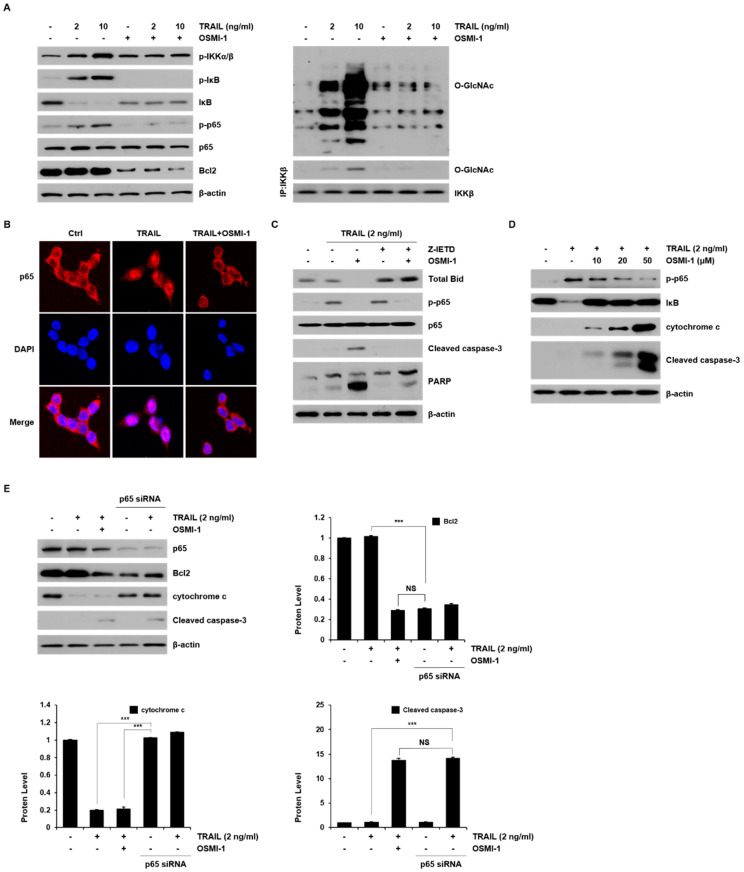
NF-κB signaling was modulated by TRAIL and OSMI-1 in HCT116. (**A**) The cells were treated with TRAIL (2 and 10 ng/mL) or combination with OSMI-1 (20 μM) for 24 h. Western blot analysis was performed to detect the levels of p-IKKα/β, p-IκB, IκB, p-p65, p65, Bcl2 (left panel), and O-GlcNAcylated protein (right panel). Total cell lysates were used for immunoprecipitation analysis of IKKβ followed by Western blot analysis with O-GlcNAc antibody (right panel). Equal amounts of the precipitate were used to assess O-GlcNAc levels. (**B**) HCT116 cells were treated with 2 ng/mL TRAIL for 30 min in the presence or absence of 20 μM OSMI-1. The localization of p65 was visualized using immunofluorescence analysis with anti-p65 (red) and the nucleus was stained with DAPI dye (blue). The merged image represents p65 nuclear transfer. (**C**) Cells were pretreated with or without Z-IETD-FMK for 1 h and then treated with TRAIL (2 ng/mL) for 24 h in the presence or absence of OSMI-1. Cell lysates were analyzed via western blot analysis using antibodies against Bid, p-p65, p65, cleaved caspase-3 and PARP. (**D**) Cells were treated with various concentrations of OSMI-1 (10, 20 and 50 μM) in the presence or absence of 2 ng/mL TRAIL for 24 h. Western blot analysis was performed to determine the levels of p-p65, IκB, cytochrome c and cleaved caspase-3. (**E**) Cells were transfected with control or p65 siRNA and then treated with TRAIL (2 ng/mL) for 24 h in the presence or absence of OSMI-1. The levels of p65, Bcl2, cytochrome c and cleaved caspase-3, were determined via western blot analysis. Data are presented as mean ± SEM, calculated from three biological replicates. *** *p*< 0.005; NS, not significant upon *t*-test.

**Figure 6 ijms-22-11073-f006:**
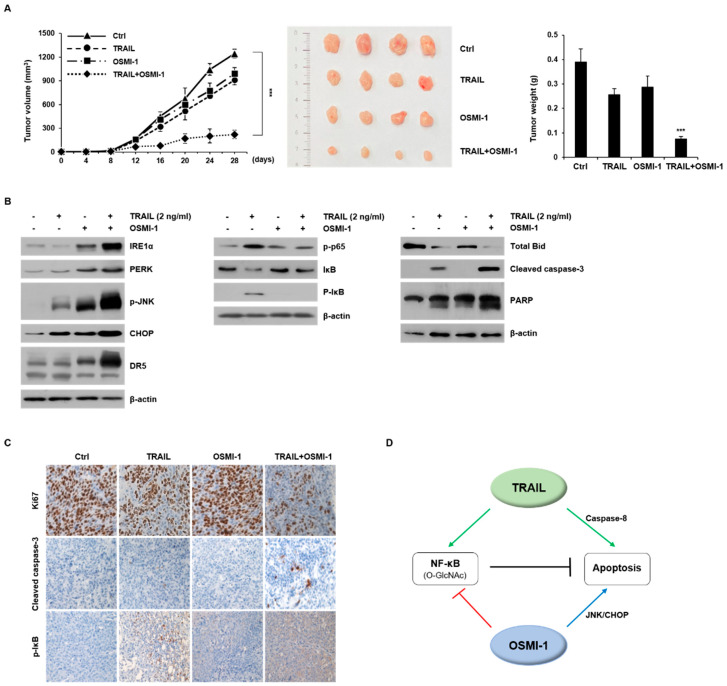
Synergic anticancer activity of combination treatment on HCT116 xenograft in nude mice. **(A)** HCT116 cells were subcutaneously inoculated in nude mice and tumor growth was monitored for 28 days. Mice were randomized into four groups. Vehicle (DMSO), TRAIL (500 μg/kg) and OSMI-1 (1 mg/kg) alone or in combination, were administrated using IV injections once every day for 3 weeks. Tumor volumes (mm^3^) were measured at the indicated time points and growth profiles are shown in the left panel. Data are shown as the mean ± SEM from three independent experiments. *** *p* < 0.005 compared with the control group. Representative images of the excised tumors derived from nude mice are shown in the middle panel. The graph on the right is the result of measuring the tumor weight at the endpoint. (**B**) Expression levels of proteins related to ER stress response (left panel), NF-κB signaling (middle panel) and apoptosis (right panel) were determined via western blot analysis using equivalent amounts of total tissue protein. **(C)** HCT116 tumors from xenograft mice were sectioned for immunohistochemical analysis using antibodies against Ki67, cleaved caspase-3 and p-IκB (magnification, x200). (**D**) Proposed signaling pathways for anticancer activity following combination treatment with TRAIL and OSMI-1. TRAIL signaling involves not only apoptosis through caspase activation but also cell growth through NF-κB. OSMI-1 blocks NF-κB signaling but activates apoptosis through ER stress activation. Arrows and bars represent activation and suppression, respectively.

## Data Availability

The data presented in this study are available on request from the corresponding author. The data are not publicly available due to privacy.

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
