# Peer review of "OSMI-1 Enhances TRAIL-Induced Apoptosis through ER Stress and NF-κB Signaling in Colon Cancer Cells"

_ijms, 2021, doi:10.3390/ijms222011073_

Round 1

Reviewer 1 Report

The study of Lee et al demonstrated a novel solution for TRAIL resistance colorectal cancer cells.  They have studied the mode of action of OGT inhibitor (OSMI-1) that sensitize the TRAIL-resistance HCT116 and HepG2 cancer cells to TRAIL-induced apoptosis but the paper raised few concerns prior to being accepted for publication. 

Major concerns

  1. Since the authors have focused on TRAIL-induced apoptosis, they have to provide evidence for the OSMI-1-related cytotoxic effect on normal epithelial cells.
  2. In Fig.2, the authors have demonstrated that Bcl2 is not affected by OSMI-1 treatment; however, in Fig.5A, the Bcl2 level sharply decreased in response to OSMI-1. How could you justify the scenario?
  3. In Fig.5E, the use of OSMI-1 in the presence of sip65 is not logical. Instead, they have to compare the effect of siP65+TRAIL and OSMI-1+TRAIL using a student's t-test.

Minor concerns

1. Authors have to follow their own standards when they use accronyms.

Ex- In the text, they have use TRAIL as well as Trail. In the Figures they use Trail. These kinds of typos reduce the overall quality of your manuscript. Please carefully check for the typos and corrected them accordingly.

2. In lines 193, 238 and 275 "μ" is missing for the concentration of OSMI-1

3. Figure arrangement especially, western blot are not impressive. You have to consider the sequence of the cell signalling as well. For example; the signals are transduced as JNK, Bcl2, Cyt.c and cas-3. Please consider the fact and arrange the bands accordingly.

4. I wonder why the authors use SD over SEM? Please justify.

5. Quality of the figures is very poor. You have to increase the resolution of the figure. 

Author Response

We would like to thank Reviewers for their comments.

The major revisions that we corrected and performed additional experiments are shown in the pdf file.

Reviewer 2 Report

- The experiments presented in figure 1 have been conducted in only one colon cancer cell line: HCT116 (HepG2 cells are hepatocellular carcinoma cells). In an attempt to generalize their result of a synergistic action of TRAIL and OSMI-1 in colon cancer cells, the experiment should be repeated in another colon cancer cell lines.

-line 195 : the authors say that “In HCT116 cells, treatment with 195 TRAIL induced 22.36±2.3% apoptosis” what about OSMI-1 alone ? please specify.

-line 219 : the authors indicate in the legend of figure 1 B that they provide “Quantitative data for three independent flow cytometry experiments” but it does not appear on the figure.

-lines 243-246 : how can the authors explain that apoptotic cell death induced by the combination of 245 TRAIL and OSMI-1 is caspase-8-dependent. it is caspase 8 dependent when osmi alone has no effect on caspase 8. Moreover why oOSMI-1 alone does not induce the cleavage of caspase 3 while it causes the release of cytochrome C ?

- the general scheme presented in Figure 6D could be more precise by displaying O-GlcNAcylation/OGT, CHOP, DR5 and JNK.

-lines 477-478: I do not understand how the authors come up with this hypothesis of an attenuation of the apoptosis signaling through upregulation of O-GlcNAcylation

-lines 483-484 : I do not understand this sentence.

- From a general point of view the quality of the figures could be improved (blurred)

- the WB and the graphs presented in figure S1 and S2 are the same ! please correct

Author Response

We would like to thank Reviewers for their comments.

The major revisions that we corrected and performed additional experiments are shown below.

Round 2

Reviewer 1 Report

The authors have dully addressed the revision points. So the current version is acceptable for publication.